# DEEP EQUILIBRIUM MULTIMODAL FUSION

## ABSTRACT

Multimodal fusion integrates the complementary information present in multiple modalities and has gained much attention recently. Existing fusion approaches exhibit three key elements for informative multimodal fusion, *i.e.*, stabilizing uni-modal signals, capturing intra- and inter-modality interactions at multi-level, and perceiving modality importance in a dynamic manner. The current fusion methods mostly suffice only one of these conditions, without considering all three aspects simultaneously. Encapsulating these ideas, in this paper, we propose a novel deep equilibrium (DEQ) method for multimodal fusion via seeking a fixed point of the dynamic multimodal fusion process and modeling feature correlations in an adaptive and recursive manner, which naturally consolidates the three key ingredients for successful multimodal fusion. Our approach encodes and stabilizes rich information within and across modalities thoroughly from low level to high level and dynamically perceives modality importance for efficacious downstream multimodal learning, and is readily pluggable to various multimodal frameworks. Extensive experiments on four well-known multimodal benchmarks, namely, BRCA, MM-IMDB, CMU-MOSI, and VQA-v2, involving a vast variety of modalities, demonstrate the superiority and generalizability of our DEQ fusion. Remarkably, our DEQ fusion consistently achieves state-of-the-art performance on these benchmarks. The code will be released.

## 1 INTRODUCTION

Humans routinely receive and process signals through interactions across multiple modalities, supporting the unique human capacity to perceive the world. With the rise and development of deep learning, there has been a steady momentum of innovation that leverages multimodal data for learning deep models (Ngiam et al., 2011; Mroueh et al., 2015; Ramachandram & Taylor, 2017). Multimodal fusion, the essence of multimodal learning, aims to integrate the information from different modalities into a unified representation, and has made great success in real-world applications, *e.g.*, sentiment analysis (Zadeh et al., 2016), multimodal classification (Arevalo et al., 2017), medical analysis (Banos et al., 2015; Wang et al., 2021), object detection (Song et al., 2015), visual question answering (Goyal et al., 2017), *etc*.

A common practice for multimodal learning is to first exploit uni-modality features, and then capitalize on multimodal fusion to combine information from all modalities, so-called *late fusion*. Uni-modal learning has progressed with advanced architectures (He et al., 2016; Vaswani et al., 2017; Liu et al., 2021), whereas multimodal fusion lags behind. In general, most fusion strategies are *static*, *i.e.*, all modality information are treated equally. This may result in generalization problems, especially for tasks involving complicated multimodal correlations. Moreover, for simple modality inputs, these static approaches might be excessive and potentially encode redundant, unstable, and even noisy information.

Revisiting recent works, we found that many of them have reached a consensus on a few key components to succeed in multimodal fusion. Joze et al. (2020) recalibrated the channel-wise features from multiple CNN streams for multimodal feature alignment. Duan et al. (2022) emphasized the importance of capturing a higher and more stable level of features to enforce better crossmodal alignment. Hou et al. (2019); Pan et al. (2020); Xue & Marculescu (2023) have found the importance of stacking multiple fusion layers to capture higher-level feature interactions. In an effort to improve the static fusion, Han et al. (2022); Wang et al. (2022a); Xue & Marculescu (2023) have devised dynamic fusion processes to account for modality importance.

We summarize these findings into three key points for successful multimodal fusion: 1) stabilizing and aligning signals from different modalities; 2) integrating interactions across modalities ranging from low level to high level; 3) dynamically perceiving the effective information and removing the redundancy from each modality. *To the best of our knowledge, there is no unified multimodal fusion framework that looks into all three aspects simultaneously.* This motivates us to develop a dynamic multimodal fusion architecture to adaptively model the cross-modality interactions from low level, middle level, to high level, making the architecture generic for various multimodal tasks.

To consolidate the above idea, we present a new deep equilibrium (DEQ) method for multimodal fusion in this paper. Our launching point is to recursively execute nonlinear projections on modality-wise features and the fused features until the equilibrium states are found. Specifically, our contributions include: *1) we seek the equilibrium state of features to jointly stabilize intra-modality representations and inter-modality interactions; 2) our method continuously applies nonlinear projections to modality-wise features and the fused features in a recursive manner. As such, the cross-modality interactions are reinforced at multiple levels for multimodal fusion; 3) we devise a* purified-then-combine *fusion mechanism by introducing a soft gating function to dynamically perceive modality-wise information and remove redundancy.* Our DEQ fusion generalizes well to various multimodal tasks on different modalities and is readily pluggable to existing multimodal frameworks.

We evaluate our DEQ fusion approach on several multimodal benchmarks built on different modalities, including medical breast invasive carcinoma PAM50 subtype classification on BRCA, image-text movie genre classification on MM-IMDB, audio-text sentiment analysis on CMU-MOSI, and image-question visual question answering on VQA-v2. Our DEQ fusion approach consistently achieves new state-of-the-art performance on all benchmarks, demonstrating the superiority of modeling stable modality information from low level to high level in a dynamic way for multimodal fusion. The related works are discussed in Appendix B.

## 2 DEEP EQUILIBRIUM FUSION

### 2.1 REVISITING DEEP EQUILIBRIUM MODEL

Our DEQ fusion is particularly built on deep equilibrium models to recursively capture intra- and inter-modality interactions for multimodal fusion. The traditional deep neural networks have finite depth and perform the backward pass through every layer. Two interesting observations are that the hidden layers tend to converge to some fixed points, and employing the same weight in each layer of the network, so-called *weight tying*, still achieves competitive results. That leads to the design principles of deep equilibrium models and the goal is to simulate an infinite depth weight-tied deep network, producing high-level and stable feature representations.

Formally, the standard DEQ (Bai et al., 2019) is formulated as a weight-tied network, and such a network with parameter $\theta$ and a depth of $L$ computes a hidden state $\mathbf{z}$ as

$$\mathbf{z}^{[j+1]} = f_\theta(\mathbf{z}^{[j]}; \mathbf{x}), \quad j = 0, \dots, L-1 \tag{1}$$

where the untransformed input $\mathbf{x}$ is injected at each layer, $\mathbf{z}^{[j]}$ is the hidden state at layer $j$ and $\mathbf{z}^{[0]} = \mathbf{0}$. As claimed in Bai et al. (2019), the core idea of DEQ is that when there are infinite layers ($L \to \infty$), the system tends to converge to an equilibrium state $\mathbf{z}^*$ such that

$$\mathbf{z}^* = f_\theta(\mathbf{z}^*; \mathbf{x}). \tag{2}$$

In practice, naively computing the equilibrium state requires excessive runtime. One convergence acceleration is to formulate Eq. (2) into a root-finding problem:

$$g_\theta(\mathbf{z}; \mathbf{x}) = f_\theta(\mathbf{z}; \mathbf{x}) - \mathbf{z}. \tag{3}$$

Some root solvers can then be applied to the residual $g_\theta$ to find the equilibrium state

$$\mathbf{z}^* = \text{RootSolver}(g_\theta; \mathbf{x}). \tag{4}$$

Instead of backpropagating through each layer, we can compute gradients analytically as

$$\frac{\partial \ell}{\partial (\cdot)} = \frac{\partial \ell}{\partial \mathbf{z}^*} \left( -J_{g_\theta}^{-1}\big|_{\mathbf{z}^*} \right) \frac{\partial f_\theta(\mathbf{z}; \mathbf{x})}{\partial (\cdot)}, \tag{5}$$

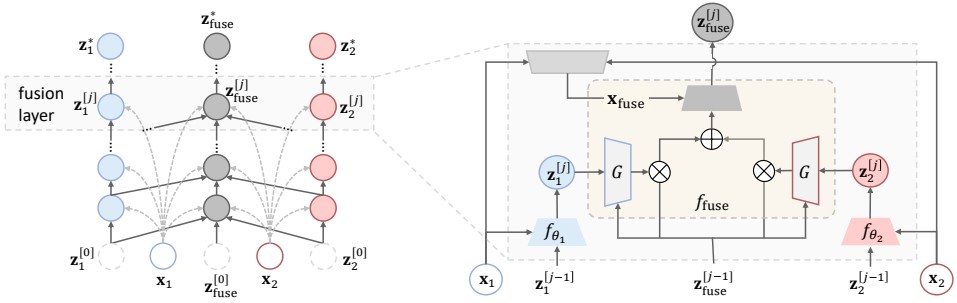

Figure 1: Our deep equilibrium fusion architecture. For simplicity, we illustrate the case where there are two modalities ($N = 2$). (Left) The fusion layer is applied in a recursive manner until the equilibrium states are reached. (Right) Each layer $j$ computes its output based on the previous iteration. $\mathbf{z}^{[j]}$ denotes the output $\mathbf{z}$ at layer $j$. The modality-wise features $\mathbf{x}_1$ and $\mathbf{x}_2$ are injected at each layer, and are combined to obtain the residual fused feature $\mathbf{x}_{\text{fuse}}$. $+$ represents summation and $\times$ denotes Hadamard product.

where $\ell = \mathcal{L}(\mathbf{z}^*, \mathbf{y})$ is a loss between $\mathbf{z}^*$ and the target $\mathbf{y}$, $J_{g_\theta}^{-1}\big|_{\mathbf{z}^*}$ is the inverse Jacobian of $g_\theta$ at $\mathbf{z}^*$. As it is expensive to compute the inverse Jacobian term, Bai et al. (2019) proposed to alternatively solve a linear system by involving a vector-Jacobian product

$$\mathbf{x} \left( J_{g_\theta}\big|_{\mathbf{z}^*} \right) + \frac{\partial \ell}{\partial \mathbf{z}^*} = \mathbf{0}. \tag{6}$$

With the formulation above, DEQ represents an infinite depth network with just one layer $f_\theta$, which converges to an equilibrium state, and can be backpropagated implicitly with a single computation.

## 2.2 Deep Equilibrium Multimodal Fusion

Next, we formally formulate our DEQ fusion method. Given a set of unimodal features $\mathbf{x} = \{\mathbf{x}_1, \mathbf{x}_2, \ldots, \mathbf{x}_N\}$ from $N$ modalities, our goal is to find a unified feature that integrates the information from all modalities. To ensure the informativeness of our final integrated feature, we first execute another nonlinear projection $f_{\theta_i}(\cdot)$ to extract higher-level information within each modality:

$$\mathbf{z}_i^{[j+1]} = f_{\theta_i}(\mathbf{z}_i^{[j]}; \mathbf{x}_i), \tag{7}$$

where $\mathbf{z}_i^{[j]}$ is the $j$-th output of the layer for modality $i$ and $\mathbf{z}_i^{[0]}$ is initialized to $\mathbf{0}$. $\mathbf{x}_i$ is the injected input feature for modality $i$. Our fusion design is flexible from the standpoint that $f_{\theta_i}(\cdot)$ can be altered arbitrarily to fit multiple modalities. In our case, $f_{\theta_i}(\cdot)$ is designed to be similar to a simple residual block (He et al., 2016). Following Bai et al. (2020), we adopt group normalization (Wu & He, 2018) instead of batch normalization (Ioffe & Szegedy, 2015) for stability. Hence, $f_{\theta_i}(\cdot)$ is formulated as

$$\hat{\mathbf{z}}_i^{[j]} = \text{ReLU}\left( \text{GroupNorm}\left( \hat{\theta}_i \mathbf{z}_i^{[j]} + \hat{\mathbf{b}}_i \right) \right)$$

$$\tilde{\mathbf{z}}_i^{[j]} = \text{GroupNorm}\left( \tilde{\theta}_i \hat{\mathbf{z}}_i^{[j]} + \mathbf{x}_i + \tilde{\mathbf{b}}_i \right) \tag{8}$$

$$f_{\theta_i}(\mathbf{z}_i^{[j]}; \mathbf{x}_i) = \text{GroupNorm}\left( \text{ReLU}\left( \tilde{\mathbf{z}}_i^{[j]} \right) \right),$$

where $\hat{\theta}_i$ and $\tilde{\theta}_i$ are the weights, $\hat{\mathbf{b}}_i$ and $\tilde{\mathbf{b}}_i$ are the bias. Given this set of modality-wise features $\{\mathbf{z}_i^{[j+1]}\}$ computed from $f_{\theta_i}(\cdot)$, where $i = 1, 2, \ldots, N$, our target is to fuse them to obtain a unified feature integrating the information from all $N$ modalities. In addition, considering that the dimension of this unified feature is limited, it necessitates dynamically selecting the most representative information from each modality-wise feature to reduce redundancy.

We propose a dynamic *purify-then-combine* fusion strategy for this purpose. We account for feature correlation between the fused feature and the modality-wise features by applying a soft gating function $G(\cdot)$, to dynamically model feature correlation via computing a weight $\boldsymbol{\alpha}_i$ for each modality:

$$\boldsymbol{\alpha}_i = G(\mathbf{z}_{\text{fuse}}^{[j]}, \mathbf{z}_i^{[j+1]})$$

$$G(\mathbf{z}_{\text{fuse}}^{[j]}, \mathbf{z}_i^{[j+1]}) = \theta_\alpha \left( \mathbf{z}_{\text{fuse}}^{[j]} + \mathbf{z}_i^{[j+1]} \right) + \mathbf{b}_\alpha, \tag{9}$$

where $\mathbf{z}_{\text{fuse}}^{[j]}$ is the fused feature from the $j$-th layer and $\mathbf{z}_{\text{fuse}}^{[0]}$ is initialized to $\mathbf{0}$. $\theta_\alpha$ and $\mathbf{b}_\alpha$ are the weight and bias. The gating function $G(\cdot)$ assigns the larger weights to parts of the fused feature that better encode the information from modality $i$. We purify the fused feature with the correlation weight for modality $i$:

$$\mathbf{z}_i' = \boldsymbol{\alpha}_i \odot \mathbf{z}_{\text{fuse}}^{[j]}, \tag{10}$$

where $\odot$ represents Hadamard product. $\mathbf{z}_i'$ could be interpreted as the significant feature purified from the fused feature that represents the information of modality $i$ from previous layers. We then combine these purified features and adopt a simplified residual block to obtain the unified feature as

$$\hat{\mathbf{z}}_{\text{fuse}} = \theta_{\text{fuse}} \cdot \sum_{i=1}^{N} \mathbf{z}_i' + \mathbf{b}_{\text{fuse}}$$
$$\mathbf{z}_{\text{fuse}}^{[j+1]} = \text{GroupNorm}\left(\text{ReLU}\left(\hat{\mathbf{z}}_{\text{fuse}} + \mathbf{x}_{\text{fuse}}\right)\right), \tag{11}$$

where $\mathbf{x}_{\text{fuse}}$ is the injected input fused feature computed from the set of modality-wise features $\{\mathbf{x}_i\}$ for $i = 1, 2, \ldots, N$, $\theta_{\text{fuse}}$ and $\mathbf{b}_{\text{fuse}}$ are the weight and bias. In shallow layers (small $j$), $\mathbf{z}_{\text{fuse}}^{[j]}$ encodes low-level modality interactions. As we continuously summarize the purified feature $\mathbf{z}_i'$, *i.e.*, $j$ gets larger and larger, $\mathbf{z}_{\text{fuse}}^{[j]}$ tends to capture higher-level modality interactions while recursively integrating low-level information from previous iterations. By doing so, the final $\mathbf{z}_{\text{fuse}}^{[\infty]}$ integrates the cross-modality interactions and correlations ranging from low level to high level. Moreover, our approach is flexible on the ways to compute the injected fused feature $\mathbf{x}_{\text{fuse}}$. In our case, we compute it with a simple weighted sum:

$$\mathbf{x}_{\text{fuse}} = \sum_{i=1}^{N} w_i \mathbf{x}_i, \tag{12}$$

where $w_i$ is a learnable weight associated with modality $i$ representing modality importance.

We denote the above-proposed fusion module in Eqs. (9) to (11) as a nonlinear function $f_{\text{fuse}}(\cdot)$ such that

$$\mathbf{z}_{\text{fuse}}^{[j+1]} = f_{\text{fuse}}(\mathbf{z}_{\text{fuse}}^{[j]}; \mathbf{x}), \tag{13}$$

where $\mathbf{x} = \{\mathbf{x}_i\}$ for $i = 1, 2, \ldots, N$ is the set of the injected modality-wise features. Ideally, a superior unified feature should capture the information from all modalities at every level and thus we progressively model modality interactions from low-level to high-level feature space. Technically, we present to recursively interchange intra- and inter-modality information until the equilibrium state is reached, to obtain such an informative unified representation in a *stable* feature space for multimodal learning. To achieve this goal, we leverage the idea of DEQs into our multimodal fusion framework. Considering $f_{\theta_i}(\cdot)$ for $i = 1, 2, \ldots, N$ and $f_{\text{fuse}}(\cdot)$ as DEQ layers, we aim to find equilibrium states such that

$$\mathbf{z}_i^* = f_{\theta_i}\left(\mathbf{z}_i^*; \mathbf{x}_i\right), \quad \mathbf{z}_{\text{fuse}}^* = f_{\text{fuse}}\left(\mathbf{z}_{\text{fuse}}^*; \mathbf{x}\right), \tag{14}$$

where $\mathbf{z}_{\text{fuse}}^*$ and $\mathbf{z}_i^*$, $i = 1, 2, \ldots, N$, are the fused feature and all unimodal features in equilibrium states respectively. Note that we also keep track of computation for each unique modality-wise feature, so that the information from different modalities can be aligned and captured at a stable level together with the fused feature. We conduct ablation studies to demonstrate the superiority of our *purify-then-combine* fusion strategy compared to other fusion variants involving DEQs. Please refer to Section 3.2 for more details.

The fixed points in Eq. (14) can be reformulated into residual functions for the root-finding problem:

$$g_{\theta_i}(\mathbf{z}_i; \mathbf{x}_i) = f_{\theta_i}(\mathbf{z}_i; \mathbf{x}_i) - \mathbf{z}_i, \tag{15}$$

$$g_{\text{fuse}}(\mathbf{z}_{\text{fuse}}; \mathbf{x}) = f_{\text{fuse}}(\mathbf{z}_{\text{fuse}}; \mathbf{x}) - \mathbf{z}_{\text{fuse}} \tag{16}$$

Finally, we can solve for features in equilibrium states via a black-box solver by minimizing the residuals $g_{\theta_i}$ for $i = 1, 2, \ldots, N$ and $g_{\text{fuse}}$:

$$\mathbf{z}^*, \mathbf{z}_{\text{fuse}}^* = \text{RootSolver}(g_\theta; g_{\text{fuse}}; \mathbf{x}), \tag{17}$$

where $\mathbf{z}^* = \{\mathbf{z}_i^*\}$ and $g_\theta = \{g_{\theta_i}\}$ for $i = 1, 2, \ldots, N$. Fig. 1 illustrates an overview of our deep equilibrium fusion architecture.

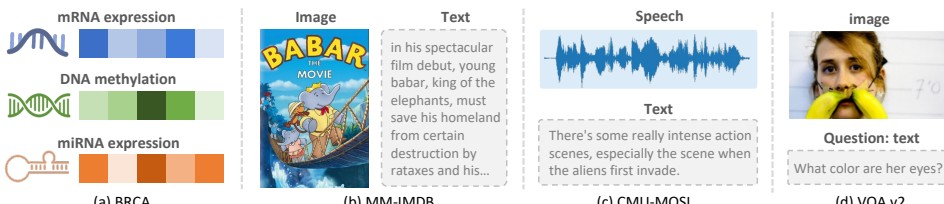

Figure 2: Data samples from the FOUR benchmarks: (a) multi-omics BRCA; (b) image-text MM-IMDB; (c) audio-text CMU-MOSI; and (d) image-question VQA-v2.

## 2.3 BACKPROPAGATION

A benefit of using DEQs compared to stacking conventional networks is that the gradients can be computed analytically without tracing through the forward pass layer-by-layer.

**Theorem 1.** *(Gradient of Deep Equilibrium Multimodal Fusion) Let $\mathbf{z}_i^*, \mathbf{z}_{\text{fuse}}^* \in \mathbb{R}^d$ for $i = 1, 2, \ldots, N$ be the equilibrium states of the modality-wise features and fused feature, and $\mathbf{y} \in \mathbb{R}^q$ be the ground-truth. Suppose we have a function $h : \mathbb{R}^d \to \mathbb{R}^q$ which is the head for some downstream tasks (e.g., classification), we can compute a loss function $\ell = \mathcal{L}(h(\mathbf{z}_{\text{fuse}}^*), \mathbf{y})$ between the prediction and the target. We can backpropagate implicitly through the unimodal features by computing the gradients with respect to $\mathbf{x}_i$ using implicit function theorem:*

$$\frac{\partial \ell}{\partial \mathbf{x}_i} = \frac{\partial \ell}{\partial \mathbf{z}_{\text{fuse}}^*} \cdot \left( -J_{g_{\text{fuse}}}^{-1} \big|_{\mathbf{z}_{\text{fuse}}^*} \right) \cdot \frac{\partial f_{\text{fuse}} \left( \mathbf{z}_{\text{fuse}}^*; \mathbf{x} \right)}{\partial \mathbf{z}_i^*} \cdot \left( -J_{g_{\theta_i}}^{-1} \big|_{\mathbf{z}_i^*} \right) \cdot \frac{\partial f_{\theta_i} \left( \mathbf{z}_i^*; \mathbf{x}_i \right)}{\partial \mathbf{x}_i}, \qquad (18)$$

*where $J_g^{-1} \big|_{\mathbf{z}}$ is the inverse Jacobian of $g$ evaluated at $\mathbf{z}$.*

The proof for Theorem 1 is provided in Appendix A. The gradients with respect to parameters of DEQ layers can be computed following Eq. (5).

## 3 EXPERIMENTS

We empirically verify the merit of our DEQ fusion on four multimodal tasks: 1) breast invasive carcinoma PAM50 subtype classification BRCA[1], associated with mRNA expression, DNA methylation, and miRNA expression data; 2) movie genre classification on MM-IMDB (Arevalo et al., 2017), which categorizes movies based on posters and text descriptions; 3) sentiment analysis on CMU-MOSI (Zadeh et al., 2016), which manually labels sentiment of video clips ranging from -3 to 3, where -3 indicates highly negative and 3 indicates highly positive; and 4) visual question answering on VQA-v2 (Goyal et al., 2017), the most commonly used large-scale VQA benchmark dataset containing human-annotated question-answer relating to images. Fig. 2 illustrates some data examples. In order to demonstrate the generalizability and plug-and-play nature of our approach, we only replace the fusion module of the existing methods and keep all the other components the same for comparison. The detailed experimental setup is demonstrated in Appendix C.

### 3.1 DISCUSSION

**BRCA.** We compare our DEQ fusion approach with several baseline fusion methods, including the current best competitor MM-Dynamics (Han et al., 2022), in Table 1. It is noticeable that the complementarity of some modalities is significant, as approximately -10% performance drop is observed without mRNA data. This also somewhat manifests the advantage of dynamic modeling to take multiple modality signals into account. Similar to our dynamic design with a soft gating function, MM-Dynamics models feature and modality informativeness dynamically for trustworthy multimodal fusion. Our DEQ fusion additionally considers intra- and inter-modality features at every level, outperforming MM-Dynamics in all evaluation metrics. Notably, our method with two modalities of mRNA and DNA methylation already attains better performance in all evaluation metrics compared to MM-Dynamics which leverages all three modalities. The above results evince the effectiveness of capturing modality interactions ranging from low level to high level in our deep equilibrium fusion design.

---

[1]BRCA can be acquired from The Cancer Genome Atlas program.

Table 1: Performance comparisons of multimodal fusion methods on BRCA benchmark. The results of baseline methods are obtained from Han et al. (2022). mR, D, and miR denote mRNA expression, DNA methylation, and miRNA expression data respectively. ↑ indicates the higher the metric the better the performance and vice versa for ↓. The best results are in bold.

|  | Modality | Acc(%)↑ | WeightedF1(%)↑ | MacroF1(%)↑ |
|---|---|---|---|---|
| GRridge (Van De Wiel et al., 2016) | mR+D+miR | 74.5±1.6 | 72.6±2.5 | 65.6±2.5 |
| GMU (Arevalo et al., 2017) | mR+D+miR | 80.0±3.9 | 79.8±5.8 | 74.6±5.8 |
| CF (Hong et al., 2020) | mR+D+miR | 81.5±0.8 | 81.5±0.9 | 77.1±0.9 |
| MOGONET (Wang et al., 2021) | mR+D+miR | 82.9±1.8 | 82.5±1.7 | 77.4±1.7 |
| TMC (Han et al., 2021) | mR+D+miR | 84.2±0.5 | 84.4±0.9 | 80.6±0.9 |
| MM-Dynamics (Han et al., 2022) | mR+D+miR | 87.7±0.3 | 88.0±0.5 | 84.5±0.5 |
| MM-Dynamics + **DEQ Fusion** | D+miR | 78.9±1.6 | 79.2±2.3 | 75.8±3.0 |
| MM-Dynamics + **DEQ Fusion** | mR+miR | 87.6±0.7 | 88.1±0.7 | 85.1±1.7 |
| MM-Dynamics + **DEQ Fusion** | mR+D | 88.7±0.7 | 89.3±0.7 | 86.9±0.9 |
| MM-Dynamics + **DEQ Fusion** | mR+D+miR | **89.1±0.7** | **89.7±0.7** | **87.6±1.0** |

Table 2: Performance comparisons of multimodal fusion methods on MM-IMDB benchmark. The results of DynMM (Xue & Marculescu, 2023), MMBT (Kiela et al., 2019), and PMF (Li et al., 2023) are obtained from their original paper. I and T denote image and text respectively.

|  | Basic Settings | Modality | MicroF1(%)↑ | MacroF1(%)↑ |
|---|---|---|---|---|
| Unimodal Image | VGGNet | I | 40.31 | 25.76 |
| Unimodal Text | Word2vec | T | 59.37 | 47.59 |
| Early Fusion | VGGNet + Word2vec | I+T | 56.00 | 49.36 |
| Late Fusion | VGGNet + Word2vec | I+T | 59.02 | 50.27 |
| DynMM | VGGNet + Word2vec | I+T | 60.35 | 51.60 |
| **DEQ Fusion** | VGGNet + Word2vec | I+T | **61.52** | **53.38** |
| Unimodal Image | ResNet152 | I | 45.65 | 29.91 |
| Unimodal Text | BERT | T | 64.81 | 58.00 |
| MMBT | ResNet152 + BERT | I+T | 66.80 | 61.60 |
| PMF-large | ViT-*large* + BERT-*large* | I+T | 66.72 | 61.66 |
| **DEQ Fusion** | VGGNet + BERT | I+T | 66.15 | 59.32 |
| **DEQ Fusion** | ResNet152 + BERT | I+T | **67.59** | **62.14** |

**MM-IMDB.** We compare our DEQ fusion strategy with various baseline fusion methods in Table 2. It is clear that text modality is more representative than image modality for this classification task, as unimodal text models exhibit significantly better performance than unimodal image models. As such, existing approaches which do not involve dynamic modeling of modality information, attain either similar performance or minor improvement compared to the unimodal text baseline. A dynamic fusion strategy is seemingly crucial to further leverage the information from the relatively weak image signal for better performance. DynMM (Xue & Marculescu, 2023) capitalizes on hard gating to select the most appropriate fusion strategy from a set of predefined operations to achieve better results. We obtain the state-of-the-art results of 61.52% and 53.38% for micro and macro F1 scores respectively on MM-IMDB benchmark with our DEQ fusion, achieving a significant improvement under the same settings against several fusion baselines. To further emphasize the generalizability and adaptability with different backbones, we additionally evaluate DEQ fusion with ResNet152 (He et al., 2016) and BERT (Devlin et al., 2018). Our DEQ fusion with ResNet152 and BERT achieves state-of-the-art results of 67.59% and 62.14% for micro and macro F1 scores respectively, surpassing MMBT (Kiela et al., 2019) which is also evaluated with the same backbones. Remarkably, our results are also better than PMF-large (Li et al., 2023) which utilizes more powerful ViT-large and BERT-large backbones.

**CMU-MOSI.** We compare our fusion approach with several baseline fusion methods, including the state-of-the-art CM-BERT (Yang et al., 2020), in Table 3. It is worth noting that BERT-based methods exhibit better performance than other baseline approaches. For instance, vanilla BERT (Devlin et al., 2018), leveraging only text modality, already surpasses other non-BERT methods which involve the utilization of all three modalities. We speculate that text modality provides more significant information for sentiment analysis task than the other two modalities. CM-BERT exploits audio modality in addition to BERT for further performance boost. Our DEQ fusion benefits from the dynamic and stable modality information modeling, and interaction exchange at every level with

Table 3: Performance comparisons of multimodal fusion methods on CMU-MOSI benchmark. The results of baseline methods are obtained from (Yang et al., 2020). T, A, and V denote text, audio, and video, respectively. Acc-$N$ denotes $N$-class accuracy.

|  | Modality | Acc-7(%)↑ | Acc-2(%)↑ | F1(%)↑ | MAE↓ | Corr↑ |
|---|---|---|---|---|---|---|
| Early Fusion LSTM | T+A+V | 33.7 | 75.3 | 75.2 | 1.023 | 0.608 |
| LRMF (Liu et al., 2018) | T+A+V | 32.8 | 76.4 | 75.7 | 0.912 | 0.668 |
| MFN (Zadeh et al., 2018a) | T+A+V | 34.1 | 77.4 | 77.3 | 0.965 | 0.632 |
| MARN (Zadeh et al., 2018b) | T+A+V | 34.7 | 77.1 | 77.0 | 0.968 | 0.625 |
| RMFN (Liang et al., 2018) | T+A+V | 38.3 | 78.4 | 78.0 | 0.922 | 0.681 |
| MFM (Tsai et al., 2018) | T+A+V | 36.2 | 78.1 | 78.1 | 0.951 | 0.662 |
| MCTN (Pham et al., 2019) | T+A+V | 35.6 | 79.3 | 79.1 | 0.909 | 0.676 |
| MulT (Tsai et al., 2019) | T+A+V | 40.0 | 83.0 | 82.8 | 0.871 | 0.698 |
| BERT (Devlin et al., 2018) | T | 41.5 | 83.2 | 82.3 | 0.784 | 0.774 |
| CM-BERT (Yang et al., 2020) | T+A | 44.9 | 84.5 | 84.5 | **0.729** | 0.791 |
| CM-BERT + **DEQ Fusion** | T+A | **46.1** | **85.4** | **85.4** | 0.737 | **0.797** |

Table 4: Performance comparisons of multimodal fusion methods on VQA-v2 benchmark. All metrics are accuracy in %.

| Basic Settings | Fusion Method | Yes/no | Number | Other | Overall |
|---|---|---|---|---|---|
| Skip-thoughts + BottomUp | Mutan | 82.40 | 42.63 | 54.85 | 63.73 |
| Skip-thoughts + BottomUp | **DEQ Fusion** | **82.91** | **45.40** | **55.70** | **64.57** |
| GloVe + BottomUp + Self-Att + Guided-Att | MCAN | 84.67 | 48.44 | 58.52 | 67.02 |
| GloVe + BottomUp + Self-Att + Guided-Att | **DEQ Fusion** | **85.17** | **49.07** | **58.69** | **67.38** |

our recursive fusion design, outperforming CM-BERT by 1.2%, 0.9%, and 0.9% in Acc7, Acc2, and F1 score, respectively.

**VQA-v2.** Our experimental results on VQA-v2 based on Mutan (Ben-Younes et al., 2017) and MCAN (Yu et al., 2019) are shown in Table 4. Mutan initializes GRU with pretrained Skip-thoughts models (Kiros et al., 2015) to process questions, whereas MCAN leverages pretrained GloVe word embeddings (Pennington et al., 2014). Both methods use bottom-up attention visual features. In addition, MCAN introduces self-attention and guided-attention units to model intra- and inter-modality interactions. Following their basic settings, we replace the fusion method with our DEQ fusion for comparison. It is noticeable that our DEQ fusion, under the same experimental settings, achieves consistent improvements over all evaluation metrics on both baselines, suggesting the superiority of our method. We are also aware of the highly performant large visual-language models, *e.g.* Radford et al. (2021); Kim et al. (2021); Li et al. (2022), in which we make comprehensive discussion and comparison with our DEQ fusion in Section 3.3.

## 3.2 ABLATION STUDIES AND ANALYSES

We conduct extensive ablation experiments and analyses to study the effectiveness of our proposed deep equilibrium fusion method from multiple perspectives. All ablation studies follow the same experimental setup as specified in Appendix C.

**Effectiveness of seeking equilibrium.** We first examine the effectiveness of computing the equilibrium state to extract and integrate stable modality information at every level. We first discard all components, *i.e.*, directly fusing with a weighted sum approach: $\mathbf{x}_{\text{fuse}} = \sum_{i=1}^{N} w_i \mathbf{x}_i$, where $w_i$ is a learnable weight associated with modality $i$. As shown in Table 5, on BRCA, this baseline fusion method obtains similar performance to Han et al. (2022). Next, we disable the recursive computation in our DEQ fusion module, *i.e.*, all $f_{\theta_i}(\cdot)$ and $f_{\text{fuse}}(\cdot)$ are only applied once without finding the equilibrium states. Since all inputs $\mathbf{z}$ are initialized to zero, this approach is equivalent to the weighted sum approach but with an additional nonlinear projection $f_{\theta_i}(\cdot)$ applied to all modality-wise features. Interestingly, introducing additional parameters without DEQ even harms performance compared to the weighted sum baseline. We additionally conduct ablation studies on MM-IMDB and CMU-MOSI, which leads to similar conclusions except that we do not observe the performance drop with additional $f_\theta$ and $f_{\text{fuse}}$ without DEQ computation (the first row in Table 7). All results demonstrate the importance of seeking the equilibrium states for multimodal fusion.

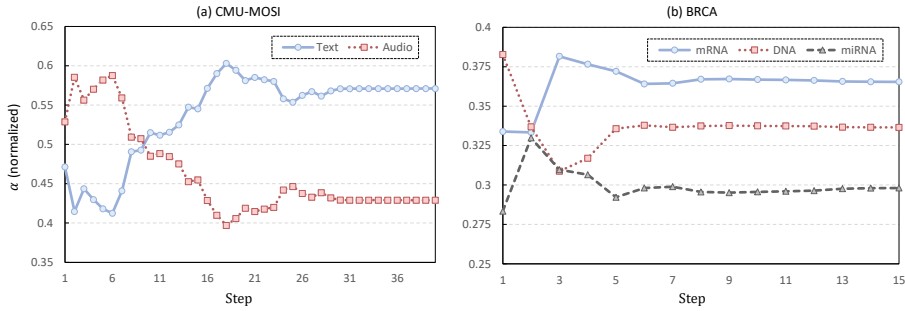

Figure 3: Plots of learned normalized $\boldsymbol{\alpha}$ over solver steps. (a) On CMU-MOSI, audio modality tends to be more important at lower levels, whereas text modality dominates over audio modality for high-level information extraction; (b) On BRCA, except for the first two solver steps, mRNA appears to be the most dominant modality.

Table 5: Ablation experiments on BRCA. $f_\theta$ represents the modality-wise nonlinear projections $f_{\theta_i}(\cdot)$ for $i = 1, 2, \ldots, N$; $f_{\text{fuse}}$ denotes the fusing function $f_{\text{fuse}}(\cdot)$; *DEQ* indicates enabling recursive DEQ computation to find the equilibrium state for the functions.

| $f_\theta$ | $f_{\text{fuse}}$ | DEQ | Acc(%)↑ | F1(%)↑ Weighted | F1(%)↑ Macro |
|---|---|---|---|---|---|
| | | | 87.6±0.4 | 87.9±0.4 | 84.3±0.8 |
| ✓ | ✓ | | 86.2±0.6 | 86.5±0.6 | 82.9±0.9 |
| ✓ | | ✓ | 88.8±0.4 | 89.4±0.4 | 87.2±0.8 |
| | ✓ | ✓ | 88.3±0.5 | 88.8±0.5 | 86.0±1.0 |
| ✓ | ✓ | ✓ | **89.1±0.7** | **89.7±0.7** | **87.6±1.0** |

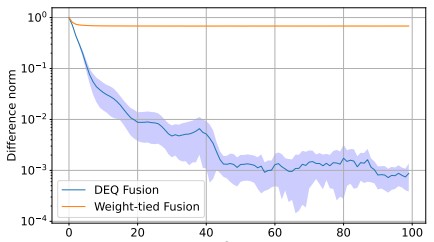

Figure 4: Plot of DEQ Fusion's convergence to equilibrium over solver steps. The shaded region indicates the 95% confidence interval computed over 10 runs.

**Different fusion variants involving DEQ.** We compare our DEQ fusion strategy against several variants involving DEQ in Table 5 and Table 7. First, we disable the *purified-then-combine* fusion strategy, *i.e.*, ablating our fusing projection $f_{\text{fuse}}(\cdot)$ by simply summating all modality-wise features: $\mathbf{z}^*_{\text{fuse}} = \sum^N_i \mathbf{z}^*_i$. Our full DEQ fusion notably improves all evaluation metrics compared to the runs without the proposed *purified-then-combine* fusion strategy. Next, we ablate all modality projections $f_{\theta_i}(\cdot)$ as identity functions by setting $\mathbf{z}^*_i = \mathbf{x}_i$. Specifically, given a set of features from $N$ modalities $\{\mathbf{x}_i\}, i = 1, 2, \ldots, N$, we set $\mathbf{z}^*_i = \mathbf{x}_i$. and proceed fusion with $f_{\text{fuse}}(\cdot)$. We notice a decline in all evaluation metrics without modality-wise nonlinear projections. These studies demonstrate that our proposed fusion variant produces the most encouraging results across all evaluation metrics.

**Impact of soft gating function.** Motivated by the success of dynamically perceiving information from modalities, we develop a soft gating function to capture the important information within each modality. We further validate the effectiveness of the proposed soft gating function $G(\cdot)$. Specifically, we set $\mathbf{z}'_i = \mathbf{z}^{[j+1]}_i$ for Eq. (10) to disable the soft gating function. As shown in Table 6 and Table 7, DEQ fusion without soft gating function causes a performance drop among all evaluation metrics. Note that since $G(\cdot)$ is a part of $f_{\text{fuse}}$, disabling $f_{\text{fuse}}$ automatically removes $G(\cdot)$. The soft gating function combined with all other components leads to the most superior result.

We additionally analyze the effectiveness of our proposed purify-then-combine fusion strategy by investigating the learned $\alpha$ values associated with different modalities in Fig. 3. On CMU-MOSI, we find that in lower solver steps, namely when extracting lower-level information, audio

Table 6: Ablation experiments of soft gating function on BRCA. $G(\cdot)$ denotes the soft gating function.

| $G(\cdot)$ | Acc(%)↑ | WeightedF1(%)↑ | MacroF1(%)↑ |
|---|---|---|---|
| | 88.4±0.8 | 89.0±0.8 | 86.1±1.1 |
| ✓ | **89.1±0.7** | **89.7±0.7** | **87.6±1.0** |

modality seems to dominate over text modality. As our DEQ fusion moves onto higher levels, text modality becomes more important. This suggests that text modality in general provides more useful information in sentiment analysis than audio modality. On BRCA, we observe that mRNA modality dominates the others, as it is always assigned with the largest $\alpha$ except for the first two solver steps. This is indeed expected, as we report in Table 1 that excluding mRNA modality leads to a significant performance drop. The observation that DNA modality is assigned with larger $\alpha$

Table 7: Ablation experiments on MM-IMDB and CMU-MOSI. "-" indicates not applicable (if $f_{\text{fuse}}$ is not used, $G(\cdot)$ is automatically disabled).

| $f_\theta$ | $f_{\text{fuse}}$ | DEQ | $G(\cdot)$ | MM-IMDB | | CMU-MOSI | | | | |
|---|---|---|---|---|---|---|---|---|---|---|
| | | | | MicroF1 | MacroF1 | Acc-7 | Acc-2 | F1 | MAE | Corr |
| | | | - | 58.76 | 49.63 | 43.3 | 83.3 | 83.2 | 0.755 | 0.786 |
| ✓ | ✓ | | ✓ | 60.73 | 52.64 | 43.0 | 83.6 | 83.6 | 0.757 | 0.787 |
| ✓ | | ✓ | - | 59.80 | 49.27 | 43.7 | 84.8 | 84.9 | 0.741 | 0.782 |
| | ✓ | ✓ | ✓ | 60.76 | 53.09 | 45.3 | 84.4 | 84.3 | 0.747 | 0.782 |
| ✓ | ✓ | ✓ | | 60.83 | 52.67 | 43.8 | 83.1 | 83.1 | 0.751 | 0.789 |
| ✓ | ✓ | ✓ | ✓ | **61.52** | **53.38** | **46.1** | **85.4** | **85.4** | **0.737** | **0.797** |

than miRNA modality also aligns with our results, as our method with mRNA+DNA reports better performance than mRNA+miRNA. These results signify that our proposed soft gating function is capable of learning and modeling modality importance, thus contributing to a more effective multimodal fusion to boost performance.

**Convergence of DEQ Fusion.** We examine the convergence of our DEQ fusion, which is an important assumption since fusion may collapse if it fails to find the equilibrium. We train a model with our DEQ fusion from scratch, and track the relative difference norm evaluated as $\|\mathbf{z}_{\text{fuse}}^{[i+1]} - \mathbf{z}_{\text{fuse}}^{[i]}\|/\|\mathbf{z}_{\text{fuse}}^{[i]}\|$ over 100 solver steps during inference. We compare it with a weight-tied fusion approach which simply iterates our fusion layer and performs backward pass layer-by-layer. Fig. 4 depicts the empirical results. It is notable that the difference norm of our DEQ fusion quickly drops below 0.01 on average within 20 solver steps, whereas the weight-tied fusion oscillates around a relatively high value. Benefiting from fixed point solvers and analytical backward pass, our DEQ fusion has much quicker and stabler convergence to the fixed point than the weight-tied approach.

## 3.3 DISCUSSION ABOUT ATTENTION-BASED FUSION

Large pre-trained models have gained popularity in multimodal learning (Lu et al., 2019; Kim et al., 2021; Li et al., 2022; Xu et al., 2023), which jointly perform feature extraction and fusion, thereby learning a task-agnostic joint representation. We refer to such approaches as attention-based fusion, which can be considered a type of mid-fusion. Despite their robust performance especially in vision-language tasks, there are several limitations that inhibit such approaches to generalize to other modalities. These methods require intricate preprocessing over input modalities such as tokenization. The lack of studies on tokenization reveals the difficulty of extending such approaches to other modalities besides image and text. Additionally, since fusion is done within the transformer modules, modifying such strategies is equivalent to adjusting the intricate backbones themselves.

DEQ fusion, as a type of late fusion approach, has several advantages. Late fusion considers feature extraction and multimodal fusion as separate processes, which neither bind to any specific input preprocessing techniques nor have any prerequisite on the unimodal feature extraction methods. As such, adapting and plugging our DEQ fusion module into existing frameworks is intuitive and straightforward. From the aforementioned perspectives, although attention-based fusion methods achieve significant results in vision-language tasks, our DEQ can be distinguished from them by benefiting from the general applicability to various modalities and backbones.

## 4 CONCLUSION

We have presented an adaptive deep equilibrium (DEQ) approach for multimodal fusion. Our approach recursively captures intra- and inter-modality feature interactions until an equilibrium state is reached, encoding cross-modal interactions ranging from low level to high level for effective downstream multimodal learning. This deep equilibrium approach can be readily pluggable into existing multimodal learning frameworks to obtain further performance gain. More remarkably, our DEQ fusion constantly achieves new state-of-the-art performances on multiple multimodal benchmarks, showing its high generalizability and extendability. A common drawback of DEQ in applications is its additional training costs for solving root-finding and uncertain computation costs during inference. Although accelerating DEQ training and inference is not a focus of this work, improving the convergence of DEQs is an important direction, which we leave as future works.

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

## A  PROOF FOR BACKPROPAGATION OF DEQ FUSION

*Proof of Theorem 1.* Our proof is similar to Bai et al. (2019). We know $\mathbf{z}_i^* = f_{\theta_i}(\mathbf{z}_i^*; \mathbf{x}_i)$ from Eq. (14), we can first differentiate two sides implicitly with respect to $\mathbf{x}_i$:

$$
\begin{aligned}
\frac{\mathrm{d}\mathbf{z}_i^*}{\mathrm{d}\mathbf{x}_i} &= \frac{\mathrm{d}f_{\theta_i}(\mathbf{z}_i^*; \mathbf{x}_i)}{\mathrm{d}\mathbf{x}_i} \\
&= \frac{\partial f_{\theta_i}(\mathbf{z}_i^*; \mathbf{x}_i)}{\partial \mathbf{x}_i} + \frac{\partial f_{\theta_i}(\mathbf{z}_i^*; \mathbf{x}_i)}{\partial \mathbf{z}_i^*} \cdot \frac{\mathrm{d}\mathbf{z}_i^*}{\mathrm{d}\mathbf{x}_i}
\end{aligned}
\tag{19}
$$

Rearranging Eq. (19), we obtain

$$
\left( \mathbf{I} - \frac{\partial f_{\theta_i}(\mathbf{z}_i^*; \mathbf{x}_i)}{\partial \mathbf{z}_i^*} \right) \frac{\mathrm{d}\mathbf{z}_i^*}{\mathrm{d}\mathbf{x}_i} = \frac{\partial f_{\theta_i}(\mathbf{z}_i^*; \mathbf{x}_i)}{\partial \mathbf{x}_i}.
\tag{20}
$$

Differentiating Eq. (15) with respect to $\mathbf{z}_i^*$, we obtain the Jacobian

$$
J_{g_{\theta_i}}|_{\mathbf{z}_i^*} = -\left( \mathbf{I} - \frac{\partial f_{\theta_i}(\mathbf{z}_i^*; \mathbf{x}_i)}{\partial \mathbf{z}_i^*} \right)
\tag{21}
$$

Therefore $\frac{\mathrm{d}\mathbf{z}_i^*}{\mathrm{d}\mathbf{x}_i} = \left( -J_{g_{\theta_i}}^{-1}|_{\mathbf{z}_i^*} \right) \cdot \frac{\partial f_{\theta_i}(\mathbf{z}_i^*; \mathbf{x}_i)}{\partial \mathbf{x}_i}$.

Similarly, we have $\mathbf{z}_{\text{fuse}}^* = f_{\text{fuse}}(\mathbf{z}_{\text{fuse}}^*; \mathbf{x}_{\text{fuse}})$ from Eq. (14). Differentiating both sides with respect to $\mathbf{z}_i^*$:

$$
\begin{aligned}
\frac{\mathrm{d}\mathbf{z}_{\text{fuse}}^*}{\mathrm{d}\mathbf{z}_i^*} &= \frac{\mathrm{d}f_{\text{fuse}}(\mathbf{z}_{\text{fuse}}^*; \mathbf{x}_{\text{fuse}})}{\mathrm{d}\mathbf{z}_i^*} \\
&= \frac{\partial f_{\text{fuse}}(\mathbf{z}_{\text{fuse}}^*; \mathbf{x}_{\text{fuse}})}{\partial \mathbf{z}_i^*} + \frac{\partial f_{\text{fuse}}(\mathbf{z}_{\text{fuse}}^*; \mathbf{x}_{\text{fuse}})}{\partial \mathbf{z}_{\text{fuse}}^*} \cdot \frac{\mathrm{d}\mathbf{z}_{\text{fuse}}^*}{\mathrm{d}\mathbf{z}_i^*}
\end{aligned}
\tag{22}
$$

Rearranging Eq. (22), we have

$$
\left( \mathbf{I} - \frac{\partial f_{\text{fuse}}(\mathbf{z}_{\text{fuse}}^*; \mathbf{x}_{\text{fuse}})}{\partial \mathbf{z}_{\text{fuse}}^*} \right) \frac{\mathrm{d}\mathbf{z}_{\text{fuse}}^*}{\mathrm{d}\mathbf{z}_i^*} = \frac{\partial f_{\text{fuse}}(\mathbf{z}_{\text{fuse}}^*; \mathbf{x}_{\text{fuse}})}{\partial \mathbf{z}_{\text{fuse}}^*}.
\tag{23}
$$

Similar to computation in Eq. (21), we have:

$$
J_{g_{\text{fuse}}}|_{\mathbf{z}_{\text{fuse}}^*} = -\left( \mathbf{I} - \frac{\partial f_{\text{fuse}}(\mathbf{z}_{\text{fuse}}^*; \mathbf{x}_{\text{fuse}})}{\partial \mathbf{z}_{\text{fuse}}^*} \right).
\tag{24}
$$

Thus $\frac{\mathrm{d}\mathbf{z}_{\text{fuse}}^*}{\mathrm{d}\mathbf{z}_i^*} = \left( -J_{g_{\text{fuse}}}^{-1}|_{\mathbf{z}_{\text{fuse}}^*} \right) \cdot \frac{\partial f_{\text{fuse}}(\mathbf{z}_{\text{fuse}}^*; \mathbf{x}_{\text{fuse}})}{\partial \mathbf{z}_{\text{fuse}}^*}$.

Finally, we can differentiate loss $\ell$ with respect to $\mathbf{x}_i$:

$$
\begin{aligned}
\frac{\partial \ell}{\partial \mathbf{x}_i} &= \frac{\partial \ell}{\partial \mathbf{z}_{\text{fuse}}^*} \cdot \frac{\mathrm{d}\mathbf{z}_{\text{fuse}}^*}{\mathrm{d}\mathbf{z}_i^*} \cdot \frac{\mathrm{d}\mathbf{z}_i^*}{\mathrm{d}\mathbf{x}_i} \\
&= \frac{\partial \ell}{\partial \mathbf{z}_{\text{fuse}}^*} \cdot \left( -J_{g_{\text{fuse}}}^{-1}|_{\mathbf{z}_{\text{fuse}}^*} \right) \cdot \frac{\partial f_{\text{fuse}}(\mathbf{z}_{\text{fuse}}^*; \mathbf{x}_{\text{fuse}})}{\partial \mathbf{z}_i^*} \cdot \left( -J_{g_{\theta_i}}^{-1}|_{\mathbf{z}_i^*} \right) \cdot \frac{\partial f_{\theta_i}(\mathbf{z}_i^*; \mathbf{x}_i)}{\partial \mathbf{x}_i}
\end{aligned}
\tag{25}
$$

$\square$

## B  RELATED WORKS

**Multimodal Fusion** aims to integrate modality-wise features into a joint representation to solve multimodal learning tasks. Early works distinguished fusion approaches into feature-level early fusion and decision-level late fusion, depending on where fusion is performed in the model (Atrey et al., 2010). Nefian et al. (2002); Xu et al. (2018) adopted early fusion approach to integrating features from multiple modalities for speech recognition and video retrieval respectively. Simonyan & Zisserman (2014) proposed to use two separate branches for spatial and temporal modalities and perform a simple late fusion for video action recognition. Alternatively, Natarajan et al. (2012) fused the outputs by computing a weighted average. Ye et al. (2012) proposed a robust late fusion using

rank minimization. More recently, with the advancement of deep learning approaches, the idea of early fusion has been extended to the concept of mid fusion, where fusion happens at multiple levels (Ramachandram & Taylor, 2017). Karpathy et al. (2014) learned the fused representation by gradually fusing across multiple fusion layers. Similarly, Vielzeuf et al. (2018) proposed a multi-layer approach for fusion by introducing a central network linking all modality-specific networks. Pérez-Rúa et al. (2019) came up with an architecture search algorithm to find the optimal fusion architecture. Hori et al. (2017); Nagrani et al. (2021) incorporated attention mechanism for multimodal fusion. Wang et al. (2020) proposed to exchange feature channels between modalities for multimodal fusion. Pan et al. (2020) introduced bilinear pooling to attention blocks, and demonstrated its superiority in capturing higher-level feature interactions by stacking multiple attention blocks for image captioning. Zhao et al. (2023) proposes a learnable pseudo-sensing module for multimodal image fusion based on physical imaging processes. Wang et al. (2022b) proposes to exchange the channels between modality-specific networks to encourage information exchange among different modalities. Duan et al. (2022) emphasized the importance of capturing a higher and more stable level of representation to enforce better cross-modality alignment for successful multimodal fusion. More recently, attention has been moved to dynamic fusion, where the most suitable fusion strategy is selected from a set of candidate operations depending on input from different modalities (Han et al., 2022; Wang et al., 2022a; Xue & Marculescu, 2023). Such dynamic approaches can adaptively capture unimodal importance, hence are more flexible to various multimodal tasks than static methods. Motivated by the success of capturing *stable* and *multi-level* feature interactions and the *dynamic* fusion designs in multimodal fusion, our work aims to integrate the information within and across modalities at multiple levels by recursively applying nonlinear projections over intra- and inter-modality features, while generalizing well to multimodal tasks involving different modalities.

Several studies have focused on transformer-based multimodal learning (Lu et al., 2019; Kim et al., 2021; Li et al., 2022; Xu et al., 2023). These approaches usually concatenate data from multiple modalities at the input level and pretrain the transformer backbone to jointly perform feature extraction and fusion via attention mechanism, and have demonstrated competitive performance especially in vision-language multimodal tasks. While these methods achieve significant improvement, our DEQ fusion, as a type of late fusion, has several benefits over such attention-based approaches. A more comprehensive discussion can be found in Section 3.3.

**Implicit Deep Learning** is a new family of deep neural networks and has grown rapidly in recent years. Traditional explicit deep models are often associated with a predefined architecture, and the backward pass is performed in reverse order through the explicit computation graphs. In contrast, implicit models compute their outputs by finding the root of some equations and analytically backpropagating through the root (Bai et al., 2020). Previous works mainly focus on designing the hidden states of implicit models. Pineda (1987) proposed an implicit backpropagation method for recurrent dynamics. Amos & Kolter (2017) proposed optimization layers to model implicit layers. Neural ODEs find the root of differentiable equations to model a recursive residual block (Chen et al., 2018). Deep equilibrium models (DEQ) find a fixed point of the underlying system via black-box solvers, and are equivalent to going through an infinite depth feed-forward network (Bai et al., 2019; 2020). These implicit deep learning approaches have demonstrated competitive performance in multiple applications while vastly reducing memory consumption, *e.g.*, generative models (Lu et al., 2021; Pokle et al., 2022), optical flow (Teed & Deng, 2020; Bai et al., 2022), graph modeling (Li et al., 2021), *etc*. Bai et al. (2021) also proposed a Jacobian regularization method to stabilize DEQs. Our work takes advantage of DEQs to adapt the number of recursion steps by finding the equilibrium state of intra- and inter-modality features jointly, and to speed up training and inference of our recursive fusion design.

## C EXPERIMENTAL SETUP

We conduct the experiments on NVIDIA Tesla V100 GPUs and use Anderson acceleration (Anderson, 1965) as the default fixed point solver for all our experiments.

**BRCA.** We experiment based on the current state-of-the-art approach (Han et al., 2022) by replacing the original concatenation fusion with our DEQ fusion. Following Han et al. (2022), the learning rate is set to 0.0001 and decays at the rate of 0.2 every 500 steps. As the dataset is relatively small, we additionally leverage dropout in fusion layer and early stopping to prevent overfitting. Jacobian

Table 8: Convergence of DEQ Fusion. The values indicate the relative difference norm computed at a given solver step.

| Dataset | step 1 | step 10 | step 20 | step 40 | step 100 |
|---|---|---|---|---|---|
| BRCA | 7.06e-1 | 3.38e-2 | 8.80e-3 | 5.18e-3 | 1.29e-3 |
| MM-IMDB | 2.86e-1 | 9.17e-4 | 7.65e-5 | 8.87e-6 | 2.17e-6 |
| CMU-MOSI | 3.09e-2 | 4.16e-7 | 6.94e-8 | 5.66e-8 | 5.66e-8 |

regularization loss with a loss weight of 20 is employed to stabilize training. We report the mean and standard deviation of the experimental results over 10 runs.

**MM-IMDB.** Our implementation and experiments on MM-IMDB are based on MultiBench (Liang et al., 2021). We follow the data split and feature extraction methods presented in Arevalo et al. (2017) for data preprocessing. Jacobian regularization loss with a loss weight of 0.1 is exploited. To further stabilize training, we additionally set a smaller learning rate of 0.0001 for our DEQ fusion module, and 0.001 for all other weights.

**CMU-MOSI.** We conduct the experiments with the state-of-the-art CM-BERT (Yang et al., 2020) by replacing the original simple addition fusion strategy with our DEQ fusion. We follow (Bai et al., 2021) and use Jacobian regularization loss with a loss weight of 0.01 to stabilize DEQ training.

**VQA-v2.** Our experiments are based on Mutan (Ben-Younes et al., 2017) and MCAN (Yu et al., 2019). All methods are trained on the train set (444k samples) and evaluated on the validation set (214k samples). Our Mutan[2] and MCAN[3] results are reproduced based on their official codebases respectively. For a fair comparison, we apply the bottom-up-attention visual features for all experiments and only use the VQA-v2 training set (disabled VisualGenome and VQA-v2 val set) for model training. Our reproduced Mutan baseline has better performance than the other reproduced version in Liang et al. (2019) (63.73% vs. 62.84% in overall accuracy) under the same settings. For MCAN, we select its "Large" model setting as our baseline.

## D    ADDITIONAL ABLATION STUDIES

In addition to the convergence ablation study on BRCA, we further examine the convergence of DEQ fusion by tracking the relative difference norm over solver steps on MM-IMDB and CMU-MOSI. The results are in Table 8. DEQ fusion successfully converges on all three benchmarks, whereas the convergence rate on MM-IMDB and CMU-MOSI is considerably faster than on BRCA.

---

[2]`https://github.com/Cadene/vqa.pytorch`
[3]`https://github.com/MILVLG/mcan-vqa`

