# OpenReview forum: "Deep Equilibrium Multimodal Fusion"
_ICLR.cc/2024/Conference — Submitted to ICLR 2024_

### Official Review · Reviewer_6XP7 · 2023-10-30

**Soundness:** 3 good
**Presentation:** 3 good
**Contribution:** 2 fair
**Rating:** 5
**Confidence:** 3

**Summary:**

This paper focus on improving multimodal fusion by developing a dynamic fusion framework adaptively model the cross-modality interactions hierarchically. The authors propose DEQ fusion which recursively executing nonlinear projections on modality-wise features and the fused features until the equilibrium states are found. Experimental results on various benchmarks can support their findings.

**Strengths:**

1. Multimodal fusion is one of the most fundamental problem in multimodal fusion, thus it is worth to design novel fusion mechanism.
2. The proposed method is novel and well motivated to me. To the best of my knowledge, most existing multmodal fusion methods tend to fuse information from multiple source in a static manner. The proposed method fuses features in a dynamic and recursive manner, which is new and interesting.
3. The proposed method are evaluated on various multimodal benchmarks, including multi-omics analysis, image-text classification, audio-text sentiment analysis and visual question answering. I appreciate the extensive experimental results. Additionally, the authors claim that the proposed DEQ fusion is readily pluggable to existing multimodal frameworks, which is very promising.

**Weaknesses:**

1. Computational cost: Though recursively fusing multimodal information is quite novel and have a chance to get better performance, I wonder if the proposed method is more time-consuming that its counterparts? It seems such comparisons is lack in the main paper and supplementary. Given this point, I encourage the authors to share more explanations about this.
2. Scalability: As the authors claim that DEQ fusion is readily pluggable to existing multimodal frameworks, I think it is deserving to further clarify how to combining DEQ fusion into existing multimodal fusion methods. For example, some pseudo code code will be very appreciated and make the proposed method easier to follow.
3. Motivation: The authors claim that DEQ fusion is an unified framework that looks into three aspects simultaneously including stabilizing and aligning multimodal signals, integrating interactions across modalities from multi-level, dynamically perceiving information. In my view, many attention-based multmodal fusion methods may also achieve the aforementioned points. In section 3.3, the authors say that DEQ fusion is not depend on any unimodal feature extraction or preprocessing methods. However, there exists some attention-based fusion methods which may also independent on unimodal feature extraction methods. For example, a simple self-attention module or MMTM[1]. Could the authors give some further clarifications?

[1] Joze, Hamid Reza Vaezi, et al. "MMTM: Multimodal transfer module for CNN fusion." Proceedings of the IEEE/CVF conference on computer vision and pattern recognition. 2020.

**Questions:**

Please refer to weakness.

---

> ### Author Response · Authors · 2023-11-14
>
> Thanks for your time reviewing our paper. Please find our responses below.
>
> > Computation cost.
>
> We have tracked the runtime of DEQ fusion on VQA-v2, the results are 0.338 and 0.405 seconds per batch without and with DEQ fusion respectively. Since the majority of the runtime is due to the unimodal feature extractors, we believe that we can tolerate this computational overhead to achieve higher performance
>
> > Scalability.
>
> Thanks for your advice. Please find the pseudo codes below written in PyTorch. Inserting our DEQ fusion requires only modifying three line of codes within the model. Also note that we will release our source codes.
>
> ```diff
>     class Model(nn.Module):
>
>         def __init__(self, ...):
>             ...
> +           self.fusion = DEQFusion(...)
>
>         def forward(self, x):
>             ...
> -           fused_feature = torch.cat([feat for feat in unimodal_feats]) # unimodal_feats contains all features extracted from unimodal encoders
> +           fused_feature = self.fusion(unimdaol_feats)
>             ...
> ```
>
> > Motivation.
>
> Thanks for pointing us to MMTM. MMTM leverages the squeeze and excitation (SE) technique to recalibrate the channel-wise features from multiple CNN streams for multimodal feature alignment. This work provides additional evidence for one of the mentioned key elements: stabilizing and aligning multimodal features. We have included MMTM in our revised paper to provide further evidence to strengthen this element.
>
> We are also aware that MMTM is also capable of capturing high-level information. Similar to other works that perform higher-level information integration, this high-level information is captured by stacking multiple fusion units. As a result, MMTM requires manual design of different architectures for different applications. Quoting from MMTM:
> >>"Although the module design is generic and could potentially be added at any level in the network hierarchy, the optimal locations and number of modules are different for each application".
>
> As such, how many fusion units are needed to obtain the optimal fused feature still remains a question for these types of works, whereas our approach is fully self-adaptive and requires minimal manual design.

---

### Official Review · Reviewer_JNZo · 2023-10-31

**Soundness:** 3 good
**Presentation:** 3 good
**Contribution:** 2 fair
**Rating:** 5
**Confidence:** 4

**Summary:**

The article presents an approach involving dynamic multi-modal fusion, introducing a weight-tied architecture to amalgamate distinct modality features and derive the unified representation simultaneously. By seeking the equilibrium state, this method facilitates the acquisition of stabilized intra-modal representations and fosters interactions across different modalities.

**Strengths:**

+ This study concentrates on the development of an innovative multi-modal fusion method, which endeavors to attain a state of equilibrium among features, markedly distinguishing itself from prior fusion processes.
+ The explanation in this article clearly and meticulously depicts its fusion architecture.

**Weaknesses:**

- The phrase "every level" in the introduction implies a comprehensive integration of cross-modality interactions throughout the multi-modal fusion process. However, given the paper’s focus on fusion of features, which is traditionally associated with late fusion, there seems to be a discrepancy. The paper apparently does not delve into early or middle fusion strategies. To reconcile this, one could interpret “every level” as referring to different stages or aspects within the late fusion process itself, although this may require clarification from the authors for a precise understanding.
-  The ablation studies conducted on the BRCA and CMU-MOSI datasets highlight the significance of the DEQ, a component not originally introduced by this work, overshadowing the impact of f_{\theta} and f_{fuse}. This raises concerns regarding the efficacy of the designed architecture in fully capitalizing on the potential for interaction among modalities. It suggests a need for further investigation and possibly a reevaluation of the architecture to ensure optimal performance.
-  The introduction categorizes three ways ‘stabilizing and aligning..., integrating..., dynamically perceiving...’ but provides limited insights, necessitating to explain the reason behind. It should be clarified that how each way contributes to better multi-modal learning. For instance, the rationale behind the need for a multi-modal model to eliminate redundancy is not clear. A thorough analysis is essential to elucidate why previous research has concentrated on these specific approaches, helping to strengthen the foundational knowledge and context for the study.
-  Furthermore, it is recommended that the authors provide a more comprehensive explanation regarding the advantages of this design in stabilizing intra-modal representations. A thorough exploration of the key difference compared to previous research, particularly in terms of enhancing stability, would enhance the reader's understanding.
-  While the proposed architecture aims to achieve stable intra-modal representations, its application is limited for it only targets to the fusion of features. This raises the question of whether the method could also encompass the learning of stable uni-modal encoders, fostering the acquisition of even more robust features. Exploring this avenue could potentially enhance the method’s applicability and effectiveness.

**Questions:**

1. It is advisable to provide detailed explanations for the dimensions of each vector and matrix utilized in the study, like $z$ in Related work.
2. Given that Multi-bench encompasses a diverse array of fusion strategies, it is crucial for the authors to specify which particular method was employed on MM-IMDB dataset.
3. The vectors $\alpha_i$ ought to be represented in bold to maintain consistency with standard mathematical notation.

---

> ### Author Response · Authors · 2023-11-14
>
> Thanks for your valuable feedback. Please find our responses below.
>
> > Confusion with the phrase "every level".
>
> As mentioned in Section 3.3, our DEQ fusion is indeed a type of late fusion. The term "every level" is thus within the late fusion process itself, where the level indicates the level of information integrated into the fused feature. When we consider the high-level information, it is mainly the product of deeper networks.
>
> > Concerns with ablation studies.
>
> Our focus is to develop a robust and generalizable multimodal fusion framework. As shown in Tables 5 and 7, naively applying DEQ to unimodal feature extractors or the fused feature leads to suboptimal results, whereas considering intra- and inter-modality features jointly (our approach) gives the best results. We would like to emphasize that our designs of $f_\theta$ and $f_{fuse}$ are not unique. One may consider employing modality-specific alternatives to further boost performance, however, this is not the focus of our work.
>
> > Further explanation on the three keys of successful multimodal fusion.
>
> We have included a further explanation in our revised paper, please find our revision for the related work section (Appendix B). Please note that these findings are suggested and backed up by several previous works.
>
> > Explanation in stabilizing representations.
>
> The principle of DEQ fusion is that the equilibrium state of the fused feature satisfies $z_{fuse}^{(i+1)}=f(z_{fuse}^{(i)})$ (slightly simplifies the notation here), which achieves stable multimodal representation while considering multi-level intra- and inter-modality feature fusion. Because it is important to ensure the *dynamic* fusion process, i.e., any modality needs to account for information from all other modalities (as suggested by [1,2,3]), we thus do not particularly emphasize unimodal feature stability.
>
> > DEQ for unimodal learning.
>
> The results for this are in Tables 5 and 7, where we evaluate DEQ applied to the unimodal encoders ($f_\theta$ with DEQ). On all three benchmarks, applying DEQ to unimodal encoders contributes to a 1-2% gain over most evaluation metrics. By jointly considering unimodal and multi-modal features, we are able to achieve optimal results.
>
> > Dimensions of the vector.
>
> The features $z$ are of the same dimension as the unimodal features $x$, such that we can compute the summation in equation 8. The dimensions of the features can be freely altered depending on the model output dimensions and different fusion module designs.
>
> > Confusion about Multi-bench.
>
> Multi-bench integrates many multimodal fusion methods and benchmarks, which is readily deployable for experiments. Our experiments including compared methods were reproduced with Multi-bench on MM-IMDB.
>
> > Vectorized $\alpha$.
>
> This is updated in the revised paper.
>
> [1] Han, Zongbo, et al. "Multimodal dynamics: Dynamical fusion for trustworthy multimodal classification." CVPR 2022.
>
> [2] Wang, Yikai, et al. "Multimodal token fusion for vision transformers." CVPR 2022.
>
> [3] Xue, Zihui, and Radu Marculescu. "Dynamic multimodal fusion." CVPR 2023.

---

> > ### Comment · Reviewer_JNZo · 2023-11-23
> >
> > Thanks for the authors' response.
> >
> > 1. Using "every level" at the begin of this tend to overclaim the feasibility of this work, and I found that this is not updated in the revision.
> > 2. The explanation about the effectiveness of DEQ is not well provided. Actually, the the results with only DEQ is required, but the authors fail to provide that.
> > 3. The three keys of successful multimodal fusion: Thanks for appending the related work in the revision, which becomes more clearly. But the better way is to revise the introduction instead of only related work.
> > 4. Explanation in stabilizing representations: With the provided explanation, this work is more like the utilization of DEQ in the multimodal settings. And as in the part *2*, the comparison with DEQ is not provided.
> > 5. Other concerns are addressed. Thanks for providing the useful respsonse.

---

> > ### Comment · Reviewer_JNZo · 2023-11-23
> >
> > Another things,
> >
> > The listed reference, dynamic multimodal fusion, is not published in CVPR 2023, but MULA workshop.

---

> > > ### Author Response · Authors · 2023-11-23
> > >
> > > Thanks for letting us know about the mistake in referencing. We have updated the reference accordingly in the revised paper.

---

> ### Author Response · Authors · 2023-11-23
>
> Thanks for following up on the discussion. We have revised our paper accordingly. Please also see our point-to-point responses below.
>
> 1. This is updated in the revised paper.
>
> 2. The relevant results are provided in the Tables 5 and 7. We will also highlight our results here and provide further explanations for your reference. On BRCA, we found that directly applying DEQ on unimodal encoders and then performing fusion with simple concatenation or summation improves the baseline accuracy to 88.8%. This result is shown in the third row of Table 5 with $f_\theta$ and DEQ. Similarly, we also experimented with applying DEQ on the fused feature (which resulted from concatenating or adding all unimodal features), which also improved the baseline accuracy to 88.3%. This result is shown in the fourth row of Table 5 with $f_{fuse}$ and DEQ. Our DEQ fusion jointly considers uni-modal and cross-modal features, leading to the most superior result of 89.1% accuracy. We conducted such ablation experiments on CMU-MOSI and MM-IMDB in Table 7 and found similar results. These experiments hopefully could demonstrate the superiority of our DEQ fusion compared with naively applying DEQ to multimodal fusion.
>
> 3. We have revised the introduction according to your advice. Please see the updated paper.
>
> 4. Please see 2 for the comparison.
>
> Thanks again for your effort and please let us know if you have further concerns.

---

### Official Review · Reviewer_qyjt · 2023-10-31

**Soundness:** 3 good
**Presentation:** 3 good
**Contribution:** 3 good
**Rating:** 5
**Confidence:** 3

**Summary:**

This paper proposes a deep equilibrium (DEQ) method for multimodal fusion by seeking a fixed point of the dynamic multimodal fusion process and modeling feature correlations in an adaptive and recursive manner.

**Strengths:**

(1)	This method innovatively combines multimodal fusion with DEQ framework to iteratively achieve multi-level multimodal fusion while retaining single-modal information
(2) 	The experiments proves the effectiveness of the method, and the ablation experiment is relatively complete. The weight visualization in Figure 3 dynamically perceives modality importance for efficacious downstream multimodal learning, which is intuitive.

**Weaknesses:**

1. The method in this paper is compared with the weight-tied method, which shows that the method in this paper can converge. This is obvious because the method optimizes fθ by the formula z* = fθ(z*,x), and does not impose such a constraint on the weight-tied method with a finite number of layers, and the weight-tied method certainly cannot converge.
2. In the original DEQ paper, DEQ is proposed for memory efficiency, and the effect is similar to that of weight-tied, and it would be better if the article gave a comparative experiment with the weight-tied method.
3. Some of the expressions in the paper are unscientific and abstract, such as the sentence on page 3:’Our fusion design is flexible from the standpoint that fθi(·) can be altered arbitrarily to fit multiple modalities. It could be better expressed as ‘Our fusion design is flexible from the standpoint that fθi(·) can be altered arbitrarily to fit multiple level features’.
4. The drawing is not intuitive.

**Questions:**

When the equilibrium state is reached, why an informative unified representation in a stable feature space for multimodal learning be obtained? What is the relationship between these two? The paper does not give proof.

---

> ### Author Response · Authors · 2023-11-14
>
> Thanks for your efforts spent on reviewing and improving our work. Please find our responses below.
>
> > Comparison against weight-tied approach.
>
> There may be some confusion here. As mentioned in Section 2.1 of our paper, the idea of finding the equilibrium is motivated by two interesting observations, which were both claimed in [1]. They found that the hidden layers of some very deep networks tend to converge to some fixed points, and employing the same weight in each layer (so-called weight-tied) also achieves similar results. From this perspective, by applying *any weight* a large number of (or infinitely many) times, the output would eventually satisfy $z_{i+1}=f_{\theta}(z_i)$. The advantages of applying DEQ are to accelerate the "infinite" forward process and memory-efficient analytical one-step backward pass.
>
> With this in mind, please note that the memory efficiency and the acceleration of the forward process come from the fixed-point solvers and analytical backpropagation, hence are not our contributions. Instead, we focus on performing robust and generalizable multimodal fusion based on DEQ. As we show in Tables 5 and 7, naively applying DEQ to unimodal feature extractors or the fused feature leads to suboptimal results, whereas considering intra- and inter-modality features jointly (our approach) gives the best results.
>
> > 'Our fusion design is flexible from the standpoint that $f_{\theta_i}$ can be altered arbitrarily to fit multiple modalities' -> 'Our fusion design is flexible from the standpoint that $f_{\theta_i}$ can be altered arbitrarily to fit multiple level features'.
>
> Our original sentence is trying to express that the design of $f_{\theta_i}$ is not unique and our design might not be optimal. One may consider altering the architecture of this module to be modality-specific to possibly further boost the performance, however, this is not our focus. Our goal instead is to propose a general framework that is applicable to most modalities and multimodal tasks. We will double check our writing to avoid such ambiguities.
>
> > When the equilibrium state is reached, why an informative unified representation in a stable feature space for multimodal learning be obtained? What is the relationship between these two?
>
> As mentioned both in Section 1 and Appendix B in our paper, various works have suggested the importance of capturing *stable* and *higher-level* representations to enforce better cross-modality for better multimodal fusion [2,3]. The principle of DEQ fusion is that the equilibrium state of the fused feature satisfies $z_{fuse}^{(i+1)}=f(z_{fuse}^{(i)})$ (slightly simplifies the notation here). By recursively applying the fusion layer, we gradually obtain higher and higher levels of feature information. The process stabilizes at the equilibrium state, as such, the unified representation at this stage captures *stable* and *multi-level* representations.
>
> The advantages of finding the equilibrium state to extract stable and multi-level unified representation are mainly evaluated empirically in our paper, rather than theoretically. Previous methods mostly extract finite levels of features, where the multi-level feature extractions are generally achieved by stacking multiple fusion blocks. Our experiments show that DEQ fusion outperforms these finite-level methods by a considerable margin, thus demonstrating the informativeness of the extracted feature from our DEQ fusion.
>
> > Confusion with drawings.
>
> We would reconsider our figures along with our captions to improve the clarity.
>
> [1] Bai, Shaojie, J. Zico Kolter, and Vladlen Koltun. "Deep equilibrium models." NeurIPS 2019.
>
> [2] Pan, Yingwei, et al. "X-linear attention networks for image captioning." CVPR 2020.
>
> [3] Duan, Jiali, et al. "Multi-modal alignment using representation codebook." CVPR 2022.

---

### Official Review · Reviewer_fFE2 · 2023-11-06

**Soundness:** 3 good
**Presentation:** 3 good
**Contribution:** 3 good
**Rating:** 6
**Confidence:** 4

**Summary:**

This paper proposed the Deep Equilibrium Multimodal Fusion (DEQ) algorithm for multimodal fusion. DEQ seeks a fixed point of the dynamic multimodal fusion process and models the feature correlations in an adaptive and recursive manner, which allows the DEQ algorithm to capture the complex dynamics of interactions between modalities. Extensive experiments demonstrated the effectiveness of DEQ.

**Strengths:**

(1) An interesting paper, the proposed DEQ method for multimodal fusion could be a new perspective in the field. By achieving equilibrium, the model could handle complex interactions between different types of data, potentially leading to better performance in tasks that require a comprehensive understanding of multimodal information. It is also a nice contribution to stability and robustness in the learning process for multimodal data.

(2) The experimental results are promising.

**Weaknesses:**

(1) DEQ models can be complex and require significant computational resources for training and inference. The search for a fixed point can sometimes lead to difficulties in convergence, especially in dynamically changing environments/contexts. There may be challenges in generalising the fixed-point approach to different types of multimodal data or applications.

(2) The paper may lack extensive evaluation against challenging applications, which is crucial to establish its real-world effectiveness. For example, I wonder how good the results of DEQ are in medical image fusion (e.g. CT, MRI, PET, etc.).

**Questions:**

Some related work is missing. For example the below [1] [2].

[1] Channel Exchanging Networks for Multimodal and Multitask Dense Image Prediction, TPAMI, 2022.
[2] 'Equivariant Multi-Modality Image Fusion' (Zhao et al, 2023).

---

> ### Author Response · Authors · 2023-11-14
>
> Thanks for your feedback and encouraging comments, please find our responses below.
>
> > Convergence and computational resources.
>
> About convergence, we have examined the convergence of DEQ fusion on three benchmarks, namely BRCA, MM-IMDB, and CMU-MOSI, involving various modalities in Table 8 of our paper. We found that our approach converges to a relatively small value on all three benchmarks, especially for common modalities like image, text, and audio, whereas the convergence on rarer medical-omics modalities is somewhat slower.
>
> Regarding the computational resources, we have tracked the runtime of DEQ fusion on VQA-v2, the results are 0.338 and 0.405 seconds per batch without and with DEQ fusion respectively. Since the majority of runtime comes from unimodal feature extractors, we believe that we can tolerate such inference speed to obtain higher performance.
>
> > More evaluations & related works.
>
> Thanks for your suggestions. We have referred to the mentioned works in our revised paper, please find our revision in the related work (Appendix B).
>
> We would again like to highlight that our DEQ fusion was evaluated on four widely used multimodal benchmarks, involving several different modalities, thus demonstrating strong generalizability. We will consider adding more benchmarks to further strengthen this point.

---

### Author Response · Authors · 2023-11-22
**Global Response**

We sincerely thank all reviewers for their thoughtful and constructive comments. In light of your valuable suggestions, we have diligently revised our paper accordingly. The modifications are detailed in the updated version of the paper for your review.

As the deadline for the discussion period is approaching, we would greatly appreciate if the reviewers could let us know whether their concerns have been adequately addressed by our responses. Additionally, please do not hesitate to request for further clarification if there is anything that remains unclear.

Once again, thanks for your invaluable efforts and insightful comments, which have been instrumental in refining our research.

---

### Meta-Review · Area_Chair_jsnH · 2023-12-06

**Metareview:**

The authors propose a new multimodal fusion method that uses deep equilibrium learning to find a unified feature that integrates the information from all modalities. The reviewers expressed concerns about a lack of clarity. I tend to agree with this assessment. For instance, I couldn't tell what the model was actually doing, at least at a high level, just from the abstract and from the comments, I had to skim through the paper. All fusion models, to some extent, capture "intra and inter-modality interactions", the contribution here appears to be the reformulation of the problem in the DEQ framework. As far as the technical contribution goes, based on a cursory read of the methods section and the comments of the reviewers, I'd judge it to be of sufficient novelty for ICLR. On the other hand, reviewers have expressed concerns about the experiments, specifically about the usefulness of the method in real world applications, the computational expense which might make larger scale experiments prohibitive, as well as the lacking ablation studies. The authors have responded to the scalability questions and have provided some further explanations on the ablation studies and on the principle of stabilizing representations. However, the performance improvements bought by the method are relatively small (1-2 F1 points) while the benchmarks do not appear to be SOTA, for instance, MFAS is missing and for MOSI all the papers are on or before 2020. So it's not really clear that what this method brings has already been accomplished, in some form or another, by other techniques.

All in all, the method presented in this paper has some novelty, however, the method isn't clearly presented, the claims are not clearly supported by the experiments and the performance improvements are not convincing, given that the baselines are older.

**Justification For Why Not Higher Score:**

The paper has some weaknesses that were not addressed in the author response, such as lack of clarity, limited theoretical contribution and insufficient experimental evidence that demonstrates the usefulness of the method.

**Justification For Why Not Lower Score:**

N/A

---

### Decision · Program_Chairs · 2024-01-16

Reject